# Damping Enhancement Using Axially Functionally Graded Porous Structure Based on Acoustic Black Hole Effect

**DOI:** 10.3390/ma12152480

**Published:** 2019-08-04

**Authors:** Weiguang Zheng, Shiming He, Rongjiang Tang, Shuilong He

**Affiliations:** School of Mechanical and Electrical Engineering, Guilin University of Electronic Technology, Guilin 541004, China

**Keywords:** acoustic black hole, functionally graded porous structure, flexural wave

## Abstract

The acoustic black hole (ABH) effect for damping flexural waves using axially functionally graded porous (FGP) structure is investigated. With proposed power-law porosity of FGP structure, ABH can be achieved and damping effect is enhanced. The physics are explained from divergent conditions of the integrated wave phase at composite ends. Numerical results show the damping effect is increased with power law index. The phenomenon is expounded by the characteristics of reflection coefficient and impedance. It indicates that increasing power law index leads to smaller wavelength along to the end, then the wave needs more oscillation cycles to travel, which leads to more energy absorption. Transient analysis for 2D FGP structure also shows the focalization and ABH effect of the flexural waves.

## 1. Introduction

Damping of vibration waves of structures is of great importance for high precision machining and measurement. Reducing wave reflection from the boundary of a structure is an effective method, especially for a uniform beam with free boundary conditions, the wave would be ideally reflected due to impedance mismatch [1]. The use of the acoustic black hole (ABH) effect as a passive control method has been studied recently in a growing interest [1,2,3,4]. By tailoring the thickness of beam with power-law profile, flexural waves in thin-walled structures gradually slow down and tend to zero in the ideal scenario, resulting in zero wave reflection at the wedge tip. This is the basic principle of ABH for vibration and acoustic control as firstly described by Mironov [5]. This technology has remarkable practical benefits since it is light weight, has a good performance, and shows appealing potential in vibration control and energy harvesting [6] due to the wave focalization effect in a confined area.

In a practical case, the perfect ABH effect would be significantly compromised from inevitable truncation at the wedge tip due to limited manufacturing capability. Krylov [7] proposed a compensated way for imperfect ABHs by covering the wedge part with a small amount of damping material, appreciable vibration damping can then be achieved. A modified thickness profile is explored to ensure and maximize the ABH effect with achievable truncation thickness by currently available manufacturing technology [8], and some promising conclusions with realistic significance are drawn. Although manufacturing imperfections are not detrimental to the ABH effect, the practical manufacture always leads to irregular extremities resulting from stress relaxation during machining [9], which limits its application in industry for some certain cases, e.g., smooth boundary and safety.

To overcome the required sophisticated thickness profile of ABH, Georgiev [10] proposed a thermal ABH by employing a shape memory polymer that is subjected to an appropriate temperature gradient. Then the storage and loss moduli vary gradually, which leads to gradually decreased phase velocity and damping efficiency is improved with the material loss factor increases. Another method for increasing boundary damping has been given by Vemula et al. [11]. The mechanism of the method is to apply gradually impedance interface to the edge of a beam, then the amplitude of vibration of composite material at free end is enlarged by the impedance gradation, which leads to large energy dissipation and attenuation of flexural wave reflection. The experiment was carried out and 60%–80% damping of energy was achieved. However, this method needs a relatively large number of different material layers, whose impedance should vary gradually to design a smooth impedance curve, which is difficult for realization and sensitivity with temperature.

Functionally graded materials (FGMs) are a new and advanced class of inhomogeneous composites, which can are made of two or more constituent materials, mixed continuously and functionally according to a given volume fraction [12]. As a result, material properties become a function of spatial position and vary gradually and smoothly in the preferred directions, which provide theoretical possibility to design a gradual impedance curve to realize the ABH effect. Recently, functionally graded porous (FGP) material has been heavily studied on its energy absorption properties, especially for vehicle safety for crushing to replace conventional honeycomb in sandwich cores [13]. The FGP also has good acoustic absorption performance as the impedance varied gradually [14]. This kind of structure is not new as lots of biological structures are functionally graded porous materials such as the bones of animals and the stalks of plants. It is obvious that such structures have the capability to dissipate vibration energy excited by running or wind. Motivated by this, the aim of the paper is to propose a theoretical possibility for FGP composite to realize the ABH effect to damping structural vibrations. Gradually varied material properties of FGP structure provide a smooth gradual impedance from stiffness to soft end, which leads to the wavelengths decrease. Then the damping of the structure would be enhanced and ABH would be effective under a given situation.

An exhaustive literature review of FGMs reveals that the majority of the studies are concentrated on free/forced vibration analysis of FG beams with material property variation along the depth of the beam. The governing equations are derived using Hamilton’s principle while employing different higher order shear deformation theories and obtained the solution to these equations using Navier solution method or finite element method (FEM) [15,16]. A few researchers have concentrated on the dynamic analysis of FG beams where the material property varies axially along the beam. Simsek et al. [17] derived the equation of motion using Lagrange’s equations and the dynamic responses of axially FG beam were solved by using the Newmark method. Shahba et al. [18,19] investigated the free vibration and stability analysis of Euler-Bernoulli and Timoshenko beams using the finite element method. The damping effects of axially FGP beams have not been studied yet.

This paper attempts to propose a new kind of ABH structure by using axially FGP structure to enhance damping effect. In Section 2, a power-law porosity of FGP structure is proposed to realize the ABH effect. The physics are explained from divergent conditions of the integrated wave phase at composite ends based on the Euler-Bernoulli hypothesis. Numerical analysis is conducted in Section 3. The finite element method is introduced to illustrate the damping effect of the composite beams for Euler-Bernoulli and Timoshenko models. The impedance and wave expansion methods are introduced to determine the reflection coefficient to illustrate the ABH effect of the composite beam. Transient analysis for 2-D FGP structure also show the focalization and ABH effect of the flexural waves. Section 4 concludes the paper.

## 2. Theoretical Analysis

A uniform beam with an axially FGP end is shown in Figure 1. The material properties of FGP beam are assumed to vary continuously through the axial direction according to the power law distribution. Then the mechanical properties of the FGP end can be expressed with the following functions [20]:(1)$$E\left(x\right)=E_{\max}{\left(1-f\left(x\right)\right)}^{2}$$
(2)$$\rho \left(x\right)=\rho_{\max}\left(1-f(x)\right)$$
where *f*(*x*), *E*(*x*), and ρ(x) are the porosity, Young’s modulus, and density at an arbitrary point through the length, and *E*_max_ and $\rho_{\max}$ are the Young’s modulus and density of the dense material. Moreover, we assume the Poisson’s ratio to be constant as already assumed in several studies.

A power-law porosity is supposed in the paper, which is:(3)$$f\left(x\right)=1-{\left(x/L_{1}\right)}^{N}$$
where *N* is the power index. The integrated wave phase resulting from the wave propagation from a certain point *x* to the free end (*x* = 0) can be described as [21]:(4)$$\Phi =\int_{0}^{x}k\left(x\right)dx$$
where $k=\frac{\omega }{c}=\sqrt[{4}]{\frac{\rho A\omega^{2}}{EI}}=\sqrt[{4}]{\frac{12\rho \omega^{2}}{Eh^{2}}}$ is wave number of flexural wave for Euler-Bernoulli beams, *E* is Young’s modulus, ρ is density, ω is angular frequency, *A* = *bh* is area of beam cross section, *b* is beam width, *h* is beam thickness, $I=\frac{bh^{3}}{12}$ is inertia moment.

Substituting Equations (1) and (2) to Equation (4), then the integrated wave phase of FGP end can be derived as:(5)$$\Phi =\sqrt[{4}]{\frac{12\omega^{2}\rho_{\max}}{h^{2}E_{\max}}}\int_{0}^{x}x^{-\frac{N}{4}}dx$$

Notice that if N≥4, Equation (5) diverges. It means the phase becomes infinite and the wave velocities approach zero; therefore, the wave never reaches the end and never reflects back, which constitutes the ABH effect. While in practical terms, the end must be truncated by the limitations of technological difficulties with manufacturing of perfect FGP around *x* = 0. The damping character of the FGP would attenuate the waves as much as possible, as when the flexural wave moves through the FGP end with power-law reduced porosity, it would asymptotically slow down, and grows in amplitude, then the damping effect would be enhanced at tip area.

## 3. Numerical Analysis

### 3.1. 1-D Beam with FGP End

To illustrate the damping effect of the composite structure, a 1D numerical model, shown in Figure 1, is presented. The parameters of the model are listed in Table 1. The uniform part of the cantilever beam is supposed with no damping. Complex Young’s modulus is introduced to express the damping characteristics of FGP end.

Frequency responses are analyzed by using finite element methods based on Euler-Bernoulli beam, shown in Figure 2, and Timoshenko beam, shown in Figure 3. In order to avoid the singularity of numerical calculation, the finite element (FE) models have a small truncation at $x_{t}=0.0006\;m$. Two nodes beam element with same length of 0.0006 m is adopted in the FEM analysis.

Figure 2 and Figure 3 show that the damping effects are enhanced with the increases of power law index *N*, which is more pronounced at high frequency. There is practically little difference between Euler-Bernoulli and Timoshenko models in the present case, as the effect of shear deformation and rotary inertia are not obvious at the analysis frequency. Then Euler-Bernoulli hypothesis can be adopted in the following paper for simplicity.

Figure 4 shows the real parts of the flexural wavelength along the *x*-axis at 8 kHz and 20 kHz. For each frequency, the wavelength decreases along *x* axis, and increasing *N* leads to smaller wavelength around FGP end. It can also be seen that as the frequency increases, the wavelength decreases also. Smaller wavelength flexural wave needs more oscillation cycles to travel to the end, which leads to more energy absorption due to the damping at the FGP end.

The vibrational state vector of the beam can be expressed as:(6)$$X={\left[\begin{array}{cccc} w & \theta & F & M \end{array}\right]}^{T}$$
where *w* is the displacement, θ is the local slope, *F* is the shear force, and *M* is the bending moment and all variables depend on the spatial coordinate *x*. The state equation of the Euler-Bernoulli beam can be written as a compact formulation:(7)$$\frac{\partial X}{\partial x}=HX$$
where
(8)$$H=\left[\begin{array}{cc} H_{1} & H_{2}\\ H_{3} & H_{4} \end{array}\right],\;H_{1}=\left[\begin{array}{cc} 0 & 1\\ 0 & 0 \end{array}\right],\;H_{2}=\left[\begin{array}{cc} 0 & 0\\ 0 & 1/EI \end{array}\right],\;H_{3}=\left[\begin{array}{cc} -\rho A\omega^{2} & 0\\ 0 & 0 \end{array}\right]H_{4}=\left[\begin{array}{cc} 0 & 0\\ -1 & 0 \end{array}\right]$$

Defining the local impedance matrix **Z** as:(9)$${\left[\begin{array}{cc} F & M \end{array}\right]}^{T}=j\omega Z{\left[\begin{array}{cc} \omega & \theta \end{array}\right]}^{T},\;\mathrm{with}\;Z=\left[\begin{array}{cc} Z_{1} & Z_{2}\\ Z_{3} & Z_{4} \end{array}\right]$$

Substituting Equation (7) to Equation (8), the Riccati equation is derived as:(10)$$\frac{\partial Z}{\partial x}=-ZH_{1}-j\omega ZH_{2}Z+\frac{H_{3}}{j\omega }+H_{4}Z$$

In the paper, the end of the beam is free, the boundary condition of Equation (10) can be expressed as Z(x)=0. Then Equation (10) can be solved by employing an adaptive Runge-Kutta-Fehlberg (RKF) method, which gives the way to compute **Z** at any coordinate. In order to avoid the singularity of numerical calculation, there is a small truncation at x=0.006m. Then the reflection matrix **R** can be derived using a standard wave approach as:(11)$$R={\left[j\omega ZE_{2}-E_{4}\right]}^{-1}\left[E_{3}-j\omega ZE_{1}\right]=\left[\begin{array}{cc} R_{1} & R_{2}\\ R_{3} & R_{4} \end{array}\right]$$
where $E=\left[\begin{array}{cc} E_{1} & E_{2}\\ E_{3} & E_{4} \end{array}\right]$, $E_{1}=\left[\begin{array}{cc} j\alpha & \alpha \\ \beta & -\beta \end{array}\right]$, $E_{2}=\left[\begin{array}{cc} -j\alpha & -\alpha \\ \beta & -\beta \end{array}\right]$, $E_{3}=\left[\begin{array}{cc} \gamma & \gamma \\ -j\delta & \delta \end{array}\right]$, $E_{4}=\left[\begin{array}{cc} \gamma & \gamma \\ j\delta & -\delta \end{array}\right]$, with $\alpha =\sqrt{\frac{k_{f}}{\rho A\omega^{3}}}$, $\beta =\sqrt{\frac{k_{f}^{3}}{\rho A\omega^{3}}}$, $\gamma =\sqrt{\frac{\rho A\omega }{k_{f}}}$, $\delta =\sqrt{\frac{\rho A\omega }{k_{f}^{3}}}$.

The scalar components, *R*_1_, *R*_2_, *R*_3_, and *R*_4_ of reflection matrix represent the reflection and coupling between evanescent and propagating flexural waves in the beam. *R*_1_ corresponds to propagating waves, *R*_4_ corresponds to evanescent waves, and *R*_2_ and *R*_3_ correspond to the coupling between these two types of waves.

In Figure 5 the reflection coefficients *R*_1_ of different models are presented as a function of frequency to illustrate the ABH effect. As the index *N* and frequency increase, the reflection coefficients decrease, which means the damping effect of the composite is enhanced. This can be an explanation of the phenomenon shown in Figure 2. It also can be seen that there are oscillations of the reflection coefficient due to the sharpness change of mechanical properties and the boundary condition in the finite length of the beam end.

Figure 6 shows the scalar impedance *Z*_1_ at 18 kHz. With the increase of *N*, the amplitudes of *Z*_1_ are attenuated and oscillate closer around the impedance $Z^{\mathrm{ref}}=\frac{1}{j\omega }E_{3}E_{1}^{-1}$ whose boundary condition of the end is non-reflection [1], which leads to less reflection as shown in Figure 5. That means the ABH effect is improved with increase of *N*, which is in accordance with the theoretical prediction. The distance between two peaks of *Z*_1_ is equal to one half of the wavelength λ(x) for uniform part of beam, while in the area of the composite FGP beam, the distance between two peaks is roughly one half of the local wavelength. We can see the wavelength decreases along *x* axis, and higher *N* leads to smaller wavelength at beam end. The phenomenon also can be seen from Figure 4. Then more oscillation cycles at the FGP ends lead to more energy absorption.

### 3.2. 2D Plate Centred with FGP Disk

For 2D application, a rectangle cantilever plate with dimensions of 0.3 × 0.15 × 0.0015 m is show in Figure 7. A central symmetry FGP disk with power-law porosity along the radius locates at the central of the plate. The symmetry plane of the FGP disk is same as the previous 1-D case with *N* = 6. The material properties are shown in Table 1. Frequency and transient analysis is investigated by FEM software COMSOL. 14,835 solid elements are meshed to ensure the computational accuracy at high frequency as shown in Figure 7. One edge of the plate is fixed at *x* = 0.

Velocity frequency responses are shown in Figure 8. The exciting point locates at (0.05, 0.075), and response point locates at (0.025, 0.075). The damping effect is enhanced with the increase of power law index N and exciting frequency, which is the same as for the 1D case. Transient analysis was then investigated. A harmonic force sin(2π×18000×t) is applied at the exciting point. The velocity contours and the maximum velocity V_max_ at different times are shown in Figure 9. We can see that bending waves propagate away from the source, then are focused to a compact region near the center of the FGP structure. Then the damping effect is enhanced and the ABH effect would be realized.

## 4. Conclusions

In this paper, an ABH effect using FGP structure is proposed and investigated. The proposed structure shows a good damping enhancement when the power-law index of porosity *N* ≥ 4, which is deduced from divergent conditions for the integrated wave phase at composite ends. 1D numerical results show the damping effect is increased with power law index and exciting frequency increase. The phenomenon is expounded by the characteristics of reflection coefficient and impedance. It indicates that increasing power law index or exciting frequency leads to smaller wavelength along to the end, then the wave needs more oscillation cycles to travel, which leads to more energy absorption. A 2D FGP composite is analyzed using commercial software COMSOL to verify the ABH effect. Velocity contours of transient analysis show the wave focalization and ABH effect.

Furthermore, the paper presents only a theoretical possibility to realize the ABH effect using FGP composite. Due to the manufacture difficulties in sample preparation, the experimental test has not been carried out. Approximate experimental results can be found in some existing literature [11].

## Figures and Tables

**Figure 1 materials-12-02480-f001:**
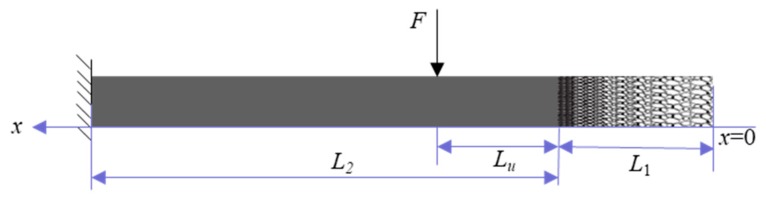
A uniform beam with functionally graded porous (FGP) end.

**Figure 2 materials-12-02480-f002:**
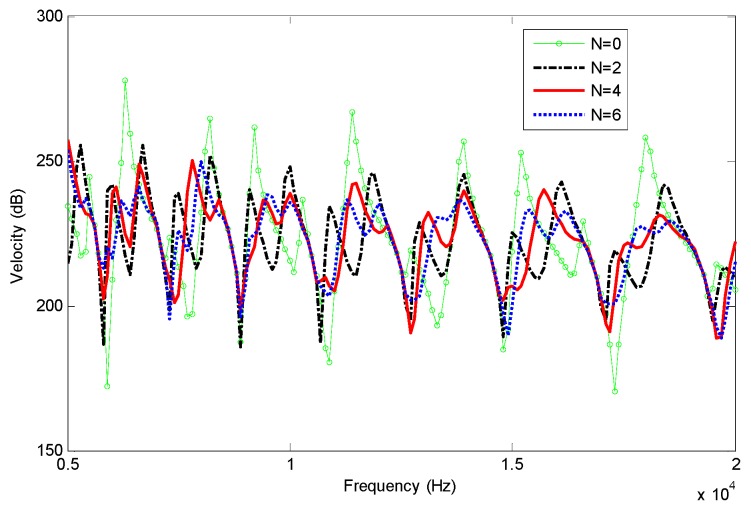
Frequency responses of the beams with FGP ends for different power law index (Euler-Bernoulli beam).

**Figure 3 materials-12-02480-f003:**
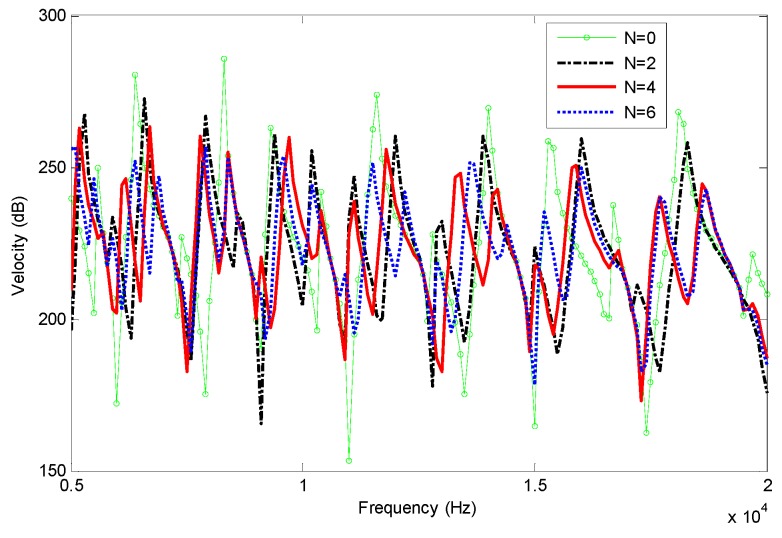
Frequency responses of the beams with FGP ends for different power law index (Timoshenko beam).

**Figure 4 materials-12-02480-f004:**
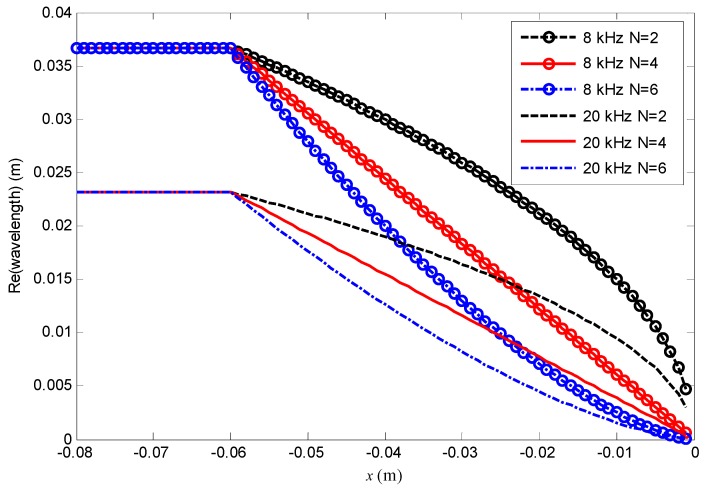
Real parts of the flexural wavelength along the *x*-axis at 8 and 20 kHz.

**Figure 5 materials-12-02480-f005:**
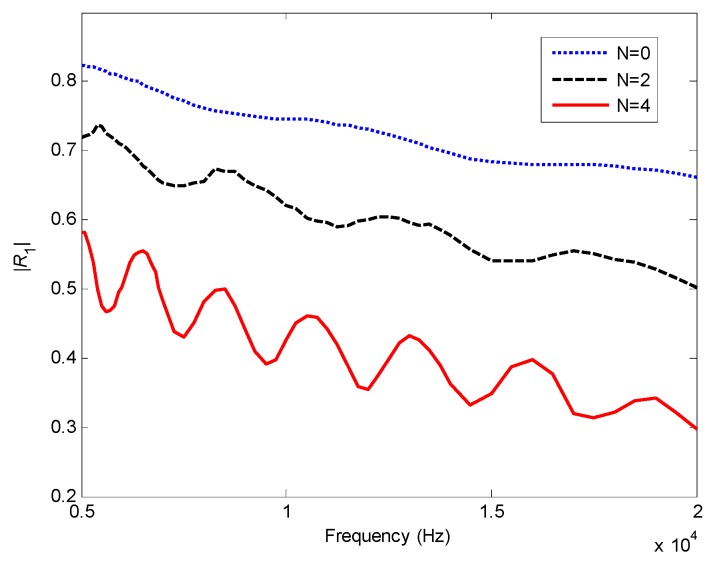
Reflection coefficients for FGP ends with different power law index *N*.

**Figure 6 materials-12-02480-f006:**
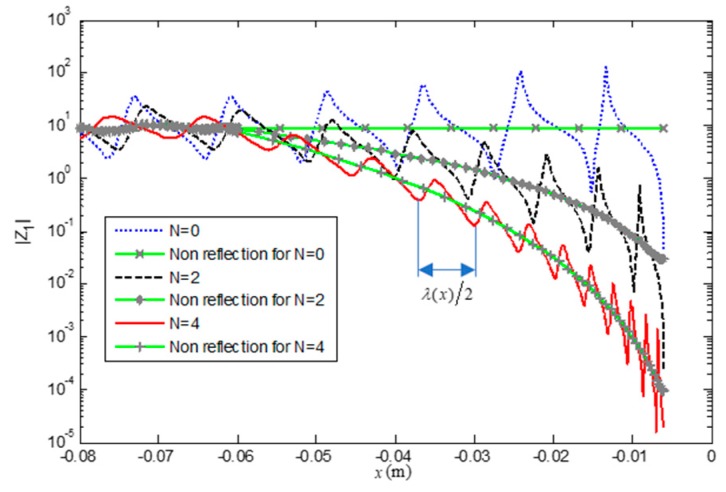
Scalar impedance *Z*_1_ along *x*-axis.

**Figure 7 materials-12-02480-f007:**
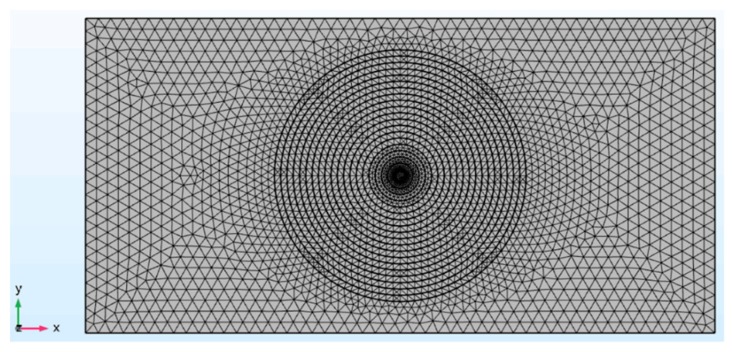
Finite element model for the 2D application.

**Figure 8 materials-12-02480-f008:**
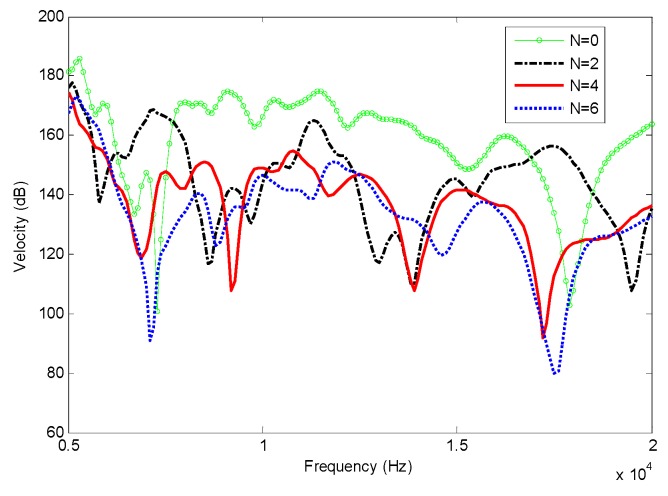
Frequency responses of 2D plate with different power law index.

**Figure 9 materials-12-02480-f009:**
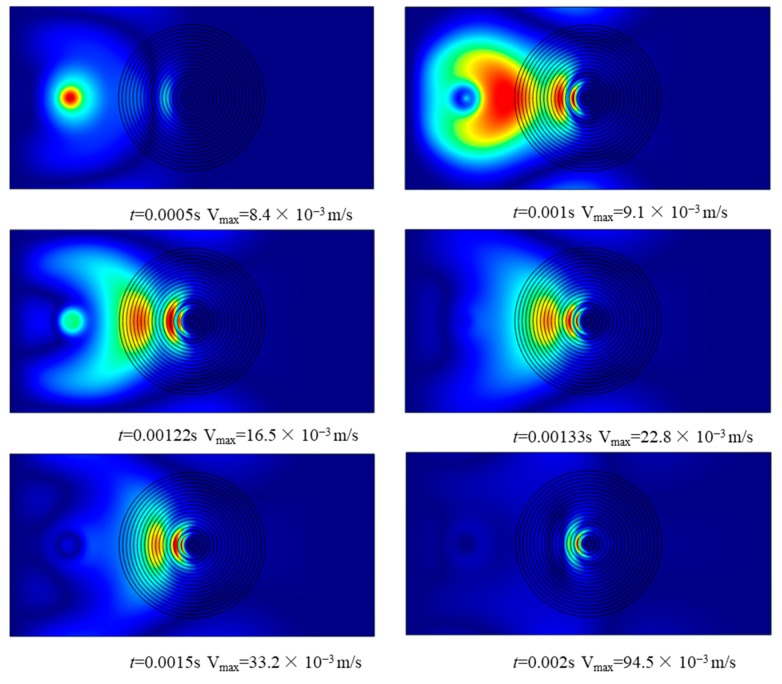
Transient analysis of 2D FGP structure.

**Table 1 materials-12-02480-t001:** Geometrical and material characteristics of the beam.

Geometrical Characteristics	Characteristics of Material
L_1_ = 0.06 m	E=100GPa
L_u_ = 0.12 m	$\rho =\rho_{\max}=6400\mathrm{kg}/m^{3}$
L_2_ = 0.24 m	η=0.05
*h* = 0.0015 m	$E_{\max}=E\left(1+i\eta \right)$
*b* = 0.0015 m

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
