# Peer review of "Damping Enhancement Using Axially Functionally Graded Porous Structure Based on Acoustic Black Hole Effect"

_materials, 2019, doi:10.3390/ma12152480_

Round 1

Reviewer 1 Report

The current manuscript requires significant improvements before considering for publication. Here are some suggestions and comments:

The writing is often so informal and does not sound scientific.

Conclusions are just list of sentences with little scientific outcomes.

Author Response

We thank you for your careful read and thoughtful comments on previous draft. We have carefully taken the comments into consideration in preparing our revision, which has resulted in a paper that is clearer, more compelling, and more scientific. All the corrections have been marked in red in the revised version.

Comments and Suggestions for Authors

The current manuscript requires significant improvements before considering for publication. Here are some suggestions and comments:

1. The writing is often so informal and does not sound scientific.

   The paper have been revised in a more scientific manner.

(1) More critical literature review is included

(2) The novelty of the manuscript is highlighted in introduction.

(3) More technical details are provided for numerical analysis.

(4)  A frequency response analysis of 2-D structure is added for a comparison of 1-D case.

(5) Some figures have been corrected.

(6) Some statements have been corrected.

2. Conclusions are just list of sentences with little scientific outcomes.

  The statements of the conclusions are revised.

Reviewer 2 Report

The idea is the paper is very interesting, I personally follow new papers on FGM for acoustic applications. So, I agree that the paper is publishable in this shape, but after a few minor revisions. First the language of the paper need slightly improvement, and second some further references are needed.

-          Line 27, deal or ideal?

-          Line 33, trunction? I think it is traction.

-          Line 38, too long to comprehend. Please split the sentence and make it more concise.

-          Line 47, strange is usually referred to something not understandable. I would easily use “new” or something more accepted.

-          Line 77, please avoid use “we …”.

-          Line 78, please remove “then”.

-          Line 24, reference would be nice.

-          Are you the first group using FGM for acoustic blackholes? If not, please reference and explain your contribution.

Author Response

We thank you for your careful read and thoughtful comments on previous draft. We have carefully taken the comments into consideration in preparing our revision, which has resulted in a paper that is clearer, more compelling, and more scientific. All the corrections have been marked in red in the revised version.

Comments and Suggestions for Authors

The idea of the paper is very interesting, I personally follow new papers on FGM for acoustic applications. So, I agree that the paper is publishable in this shape, but after a few minor revisions. First the language of the paper need slightly improvement, and second some further references are needed.

The language are improved, and more critical references are added.

1.   Line 27, deal or ideal?

      It has been revised.

2.   Line 33, trunction? I think it is traction.

      “truncation” is correct here.

3.   Line 38, too long to comprehend. Please split the sentence and make it more concise.

We have shorten the statements.

4.  Line 47, strange is usually referred to something not understandable. I would easily use “new” or something more accepted.

     It has been revised.

5.  Line 77, please avoid use “we …”.

    It has been revised.

6.  Line 78, please remove “then”.

  It has been revised.

7.  Line 24, reference would be nice.

     It has been revised.

8.  Are you the first group using FGM for acoustic blackholes? If not, please reference and explain your contribution.

    There are some application in sound absorption by using FGP see ref. [14]. We have added some critical reference to realize ABH effect [10]. Motived by the references, the concept of ABH using FGP composite is firstly proposed in the paper.

Reviewer 3 Report

Currently the novelty of the manuscript should be highlighted, it is not clearly mentioned in the text;

The structure of the "Introduction" should be modified. More critical literature review is required. 

Not enough references have been cited and commented upon;

More discussion seems necessary before 'Conclusions";

Conclusions should support the statements of the research.

More technical details should be provided for numerical analysis;

Currently there is no experimental verification in the research this should be addressed in the conclusions also in "Introduction".

Author Response

We thank you for your careful read and thoughtful comments on previous draft. We have carefully taken the comments into consideration in preparing our revision, which has resulted in a paper that is clearer, more compelling, and more scientific. All the corrections have been marked in red in the revised version.

Comments and Suggestions for Authors

1. Currently the novelty of the manuscript should be highlighted, it is not clearly mentioned in the text;

  The introduction and conclusion parts of the paper are revised to highlighted the novelty of the manuscript.

2. The structure of the "Introduction" should be modified. More critical literature review is required. 

   The introduction part are modified.

3. Not enough references have been cited and commented upon;

More references are cited and commented. 

4. More discussion seems necessary before 'Conclusions";

(1)  A frequency response analysis of 2-D structure is added for a comparison of 1-D case.

(2) Discussion about the ABH effects for different power law index and frequency is revised.

5. Conclusions should support the statements of the research.

   The conclusions is carefully considered and support the statements of the research

6. More technical details should be provided for numerical analysis;

   Details for numerical analysis have been provided. 

7. Currently there is no experimental verification in the research this should be addressed in the conclusions also in "Introduction".

 The paper is mainly verified by FEM. In conclusions, we added the statements Furthermore, the paper presents only a theoretical possibility to realize the ABH effect using FGP composite. Due to the manufacture difficulties in sample preparation, the experimental test has not been carried out. Approximate experimental results can be found in some existing literature [11]. In Introduction, Ref. [11] is commented, and some experimental results can be roughly verify the paper.

Round 2

Reviewer 1 Report

No, suggestions.

Reviewer 3 Report

I believe that the authors have addressed all the comments and the manuscript has been significantly improved and now warrants publication in Material.